Manuscript prepared for Atmos. Chem. Phys.
with version 2015/04/24 7.83 Copernicus papers of the LaTeX class copernicus.cls.
Date: 2 February 2017

# Technical Note: Simultaneous fully dynamic characterization of multiple input-output relationships in climate models

Ben Kravitz[1], Douglas G. MacMartin[2,3], Philip J. Rasch[1], and Hailong Wang[1]

[1]Atmospheric Sciences and Global Change Division, Pacific Northwest National Laboratory, Richland, WA, USA.
[2]Department of Computing and Mathematical Sciences, California Institute of Technology, Pasadena, CA, USA.
[3]Sibley School of Mechanical and Aerospace Engineering, Cornell University, Ithaca, NY, USA.

*Correspondence to:* Ben Kravitz, Atmospheric Sciences and Global Change Division, Pacific Northwest National Laboratory, P.O. Box 999, MSIN K9-30, Richland, WA 99352, USA (ben.kravitz@pnnl.gov).

**Abstract.** We introduce *system identification* techniques to climate science wherein multiple dynamic input-output relationships can be simultaneously characterized in a single simulation. This method, involving multiple small perturbations (in space and time) of an input field while monitoring output fields to quantify responses, allows for identification of different timescales of climate

response to forcing without substantially pushing the climate far away from a steady state. We use this technique to determine the steady state responses of low cloud fraction and latent heat flux to heating perturbations over 22 regions spanning Earth's oceans. We show that the response characteristics are similar to those of step-change simulations, but in this new method, the responses for 22 regions can be characterized simultaneously. Furthermore, we can estimate the timescale over

which the steady state response emerges. The proposed methodology could be useful for a wide variety of purposes in climate science, including characterization of teleconnections and uncertainty quantification to identify the effects of climate model tuning parameters.

## 1   Introduction

Understanding the response of climate models to perturbations is one of the core questions in climate

science. Some of the emergent behaviors in climate model response, particularly on small temporal and spatial scales, can be challenging to interpret. This is in part due to issues with low signal-to-noise ratios (SNRs), climate system nonlinearities, and other far-field effects.

Simulations to understand climate response frequently use abrupt or "step" changes in an exogenous input field (e.g., an abrupt increase in the $CO_2$ concentration) or "ramp" changes (e.g., a 1%

increase in the $CO_2$ concentration each year). However, in climate model simulations, the input signal can be chosen based on criteria specific to the intended goal of the simulation. Any input signal

will result in a portion of the response that is linear and a portion that is nonlinear, and increasing the magnitude of the input has the potential to amplify nonlinearities. Avoiding this prospect requires multiple ensemble members or longer simulations to increase SNR, which becomes quite expensive

if one wishes to assess multiple perturbations (e.g., changes in multiple geographical regions). As we discuss in the following section, many types of simulations that are commonly employed in climate science to investigate climate model response suffer from issues associated with this tradeoff. Moreover, they are not designed to investigate multiple input-output relationships simultaneously, necessitating larger computational cost to investigate complex systems.

Here we introduce a method of identifying input-output relationships in climate models for multiple simultaneous perturbations with relatively low computational expense and without the typical difficulties in signal detection arising from strong forcing and nonlinearities that are found in other methods commonly used in climate science. An additional advantage of this method is that it is dynamic (characterizes a range of timescales) rather than static (only characterizes the steady state

response). This methodology is commonly called *system identification* in engineering fields (Pintelon and Schoukens, 2012). In subsequent sections, we discuss the process of system identification, its utility as compared to other commonly used methods of assessing climate system behavior, and potential implications for understanding far-field effects.

## 2    System Identification

*System identification* refers to the process of using input and output time series to understand the (possibly dynamic) relationship between them. For example, if one wants to understand the climate response to a change in the $CO_2$ concentration, one can create a time-varying series of $CO_2$ concentrations, insert it into a climate model, and analyze various output fields (like global mean temperature or cloud fraction) to understand how those output fields change in response to the in-

puts. Characterizing input-output relationships should be done in a way that depends on the system to be characterized and on the objectives of the analysis. One can choose the frequency content of the input signal that one uses to characterize the system.

   Any system will respond differently to input signals at different frequencies (that is, the input-output relationship is in general dynamic). However, for many real-world systems, there is some

sufficiently low frequency for which the response is approximately the same as the equilibrium or steady-state response; this is called the *quasi-static* regime. A conceptually simple approach to characterizing a single input-output relationship in the quasi-static regime is a step response simulation, in which the input is abruptly changed. We discuss step response simulations in more detail in Section 2.1 below.

If one is interested in estimating the fully dynamic response (i.e., on different timescales over which the response varies), then the signal energy needs to be injected over a range of frequencies.

There are several strategies for accomplishing this (e.g., Kravitz et al., 2016b). A sinusoidal input puts the maximum possible power into a single frequency. However, characterizing the system at multiple frequencies requires multiple sinusoids. One could also use a wider band of frequencies (e.g., band-pass filtered white noise) at the cost of input power.

If one wishes to characterize multiple input-output relationships simultaneously (i.e., not by conducting one simulation for each input), then the different input signals need to be chosen uncorrelated from each other; this is clearly not possible with step inputs. As an example, one could choose multiple sinusoids with non-equal frequencies, which is effective if one wishes to characterize quasi-static behavior for all of the input-output relationships. Careful choice of frequency may be necessary because any nonlinearities will excite oscillations in the output that are higher harmonics of the input (e.g., an input signal of 10 Hz will result in output only at 10 Hz if the system is linear, but also at 20, 30, 40, ... Hz if there are nonlinearities); as such, it is often useful to choose non-commensurate frequencies to quantify the magnitude of the nonlinear portion of the response. If there are multiple input variables, and if one is interested in an estimate of the fully dynamic system, the input signals all need to contain broad frequency content but must be mutually uncorrelated. This is the case on which this manuscript focuses; we discuss this in more detail in Section 2.2 below.

## 2.1 Step Response Simulations

Step response simulations, in which a sustained perturbation is applied to the system, are common in climate science (e.g., Good et al., 2013). An example is the abrupt4xCO2 simulation (illustrated in Figure 1a) in which the $CO_2$ concentration is abruptly quadrupled from its preindustrial value, and the model behavior then evolves over time. The abrupt4xCO2 simulation is a standard experiment in the Coupled Model Intercomparison Project Phase 5 (CMIP5; Taylor et al., 2012). These sorts of simulations are easy to perform, and they often have high SNRs, which makes for relatively straightforward analysis.

However, there are several features of such step response simulations which, depending on the situation, may be detrimental to analysis. As described previously, if one wishes to evaluate the steady-state or quasi-static behavior, step response simulations are often an excellent tool. However, they are not well suited for evaluating fully dynamic behavior. This can be seen through the frequency decomposition of a step function (calculated via Laplace transform):

$$H(s) = \frac{1}{s} \tag{1}$$

where $s = i\omega$, and $\omega$ is (angular) frequency. At high frequencies, the input signal does not contain much energy, so unless there is sufficient amplification by the system at these frequencies, evaluating transient or short-term behavior is difficult and may require averaging multiple ensemble members.

Moreover, depending upon the magnitude of the step change and the details of the dynamical system, the resulting climate can be pushed relatively far away from the initial climate. This has the

potential to exacerbate nonlinearities in the climate response. As can be seen in Figure 1b, doubling the estimated effective radiative forcing (the $y$-intercept) or the estimated equilibrium temperature (the $x$-intercept) for an abrupt doubling of the $CO_2$ concentration does not give the same answer as for an abrupt4xCO2 response. In Figure 1, differences between these estimated quantities are 4 and 10%, respectively. In some circumstances, this may be an acceptable margin of error, and it may not be in others.

The departure from linearity can be seen more clearly when calculating the amount of heat added to the system from these runs. The total heat accumulated through a given year $n$ can be estimated by

$$\Delta Q_n = \sum_{i=1}^{n} \Delta R_i \cdot 86400 \cdot 365 \cdot A \tag{2}$$

where $\Delta R_i$ is the net top-of-atmosphere (TOA) radiative flux imbalance (W m$^{-2}$) in year $i$ that is the result of the step function perturbation, and $A$ is Earth's surface area (m$^2$). These quantities are plotted in Figure 1c for abrupt4xCO2 and two times abrupt2xCO2. Although nonlinearities account for approximately 1% of the difference between these two plotted quantities, the net difference represents a substantial amount of heat.

## 2.2 Generating multiple uncorrelated broadband input signals

Although useful for certain applications, step response simulations are not ideal for characterizing system behavior at all frequencies, and one cannot attribute the effects of multiple simultaneous step perturbations unless the responses to different inputs are independent. Simultaneously characterizing multiple dynamic input-output relationships requires constructing a set of inputs that have broad frequency content and are mutually uncorrelated.

The frequency content of the input signals is a choice, depending on the behavior in which one is interested. For example, if one cares about teleconnections on sub-annual timescales, then one could choose high-pass filtered white noise with a cutoff frequency corresponding to a timescale of one year. Similarly, if one were not interested in the high frequency response (which may also be more difficult to distinguish from internal variability), one could choose a set of low-pass filtered white noise signals. If one wishes to avoid the issue of adding substantial amounts of heat to the climate system (as was described in the previous section), one could ensure that the input signals are chosen to have zero mean; this condition is automatically satisfied by white noise.

Once these signals are generated, the next step is to ensure that they are mutually uncorrelated. This is accomplished by the Gram-Schmidt process. Let $\{v_i\}_{i=1}^{n}$ be a set of $n$ generated input signals with the appropriate frequency content for the problem of interest. Beginning with the first signal, and for each subsequent signal, one subtracts off any correlation with the previous signals to obtain

the set $\{u_i\}_{i=1}^n$. Mathematically, this is represented by

$$
\begin{aligned}
u_1 &= v_1 \\
u_2 &= v_2 - \text{proj}_{u_1}(v_2) \\
u_3 &= v_3 - \text{proj}_{u_1}(v_3) - \text{proj}_{u_2}(v_3) \\
&\quad \dots
\end{aligned}
\tag{3}
$$

where

$$
\text{proj}_u(v) = \frac{\langle v, u \rangle}{\langle u, u \rangle} u
\tag{4}
$$

and $\langle , \rangle$ represents an inner product (straightforward for discrete time; a common representation of an inner product in continuous time is an integral, as in Equation 6 below). The final stage is renormalization, where the final signals to be used $\{e_i\}_{i=1}^n$ are given by

$$
e_i = \frac{u_i}{\|u_i\|}
\tag{5}
$$

Each of these signals in the set $\{e_i\}$ is uncorrelated, has a maximum root-mean-square (amplitude in the $\ell^2$ norm) of 1 (these can be scaled as needed), and all signals have the same frequency content as the original signals $\{v_i\}$.

We define the signals to be uncorrelated (orthogonal) if

$$
\int_0^T e_i(t)e_j(t)\, dt = 0
\tag{6}
$$

for $i \neq j$, where $T$ is the length of the signals (summation can be used instead of integration for discrete systems). This criterion will ensure minimal cross-talk between the response patterns excited by individual signals, but only in the quasi-static regime where there is little dependence upon frequency. Ensuring minimal cross-talk on the fully dynamic range of frequencies would require the criterion

$$
\int_0^{T-\tau} e_i(t)e_j(t+\tau)\, dt = 0 \qquad (\forall \tau \leq t)
\tag{7}
$$

for $i \neq j$. This additional criterion accounts for lag effects (quantified as a phase shift between the input and output fields) over a range of timescales on which processes operate. As will be discussed later, for the variables analyzed here, the quasi-static state is reached relatively early in the simulations, so lag effects are not of substantive concern.

## 2.3  Climate model simulations

Once the signals are generated, the procedure is straightforward. In a climate model simulation, one modifies each of the input fields by perturbing them according to their corresponding input

signals (here, adding the input signals to the fields; see Sections 3.2 and 3.3 below for more concrete examples). After the simulation is completed, an estimate of the quasi-static sensitivity of the output to changes in the input can be obtained by projecting any time series from the resulting simulation ($U$) onto one of the original signals $a_i$ via

$$P_{U,i} = \frac{\langle a_i, U \rangle}{\langle a_i, a_i \rangle} \tag{8}$$

For example, if $a_i$ is a signal describing perturbations to sea surface temperatures in the Pacific Ocean (K), and if $U$ is a timeseries of maps of total cloud cover (%), then $P_{U,i}$ will be a two-dimensional field with units $\% \, K^{-1}$. If the response is truly static (independent of frequency), then this projection gives the best estimate of the sensitivity. Estimates of the dynamic (frequency-dependent) response

can be obtained by first band-pass filtering both the input and output signal prior to the projection in Equation 8. By choosing different filters, one can identify how the input-output relationship depends on frequency, and in particular identify the time scale at which the response is quasi-static (approximately independent of frequency). This is the procedure followed in Section 3.3. Using an appropriate low-pass filter to focus on the quasi-static regime gives a better estimate of the input-

output relationship than using Equation 8 directly on the full time series.

## 3    Demonstration of the Technique

### 3.1    Experimental Design

To apply perturbations, we need to decide on what to perturb and what to analyze. Here the perturbations applied are to air temperature near the surface over 22 regions covering the world's oceans

(Figure 2), as well as the Mediterranean Sea, chosen for its fairly large area and potential climatic importance (e.g., Paeth et al., 2016). This choice of input is an idealized representation of a change in heat flux at the surface that might be due to a change in surface sensible heat flux (through some perturbation we do not specify here) or through a surface radiative flux change like what might be produced by marine cloud brightening (Latham et al., 2012). We then analyze the effects of these

*multiple simultaneous uncorrelated broadband* perturbations on low cloud cover and latent heat flux in climate model simulations. All simulations were conducted using the fully coupled Community Earth System Model (CESM) version 1.2.0 (Hurrell et al., 2013) with $2°$ horizontal atmospheric resolution and approximately $1°$ resolution in the ocean. All simulations were conducted against a preindustrial control background.

The first step is to generate the sequences that will be used to guide model perturbations. We are *a priori* uncertain as to the timescales on which the chosen outputs will respond. As such, the most agnostic choice for the input signals is white noise, which has zero mean and content at all frequencies. (Note that because this procedure must be discretized, any input signal is effectively low-pass filtered, where the highest frequency contained in the signal corresponds to the model

timestep, which is 30 minutes.) For the purposes of this illustration, we choose to low-pass filter the white noise signals with a cutoff frequency of one week. This choice of cutoff frequency minimizes the response excited at diurnal or weekly timescales, which is a plausible choice if one wishes to characterize climatological response and eschew meteorological response.

The next step is to choose the update rate, i.e., how often the perturbation to the climate system is changed. By the Nyquist limit, the slowest possible update rate is twice the filter cutoff frequency, i.e., half a week. The difference between the cutoff frequency and the update rate is analogous to the problem of aliasing in sampling a sinusoidal curve: the sampling frequency can be different from the frequency of the actual sine wave, but obtaining an accurate fit of the sinusoid is easier if the curve is sampled more frequently, and there is a mathematical lower limit as to the minimum number of points required to obtain that fit. Here we choose the update rate to be every model day, wherein the perturbation is maintained for an entire model day. Because of practical limitations, all simulations in this study are conducted for 20 years. For all analyses of the system identification simulation in this study, we do not explicitly consider response times longer than one year. Beyond one year, there are too few points to average to obtain adequate estimates of the signal above the estimated error.

We generate 22 uncorrelated sequences as described earlier and use these sequences to perturb temperature in the lowest model layer over each of the 22 regions in Figure 2. The sequences are normalized so that values range between -1 K and 1 K, with a median magnitude of 0.3 K. Because the sequences were generated from white noise, they have a mean value of 0 K. Figure 3 shows an example of one of the 22 sequences for both the time domain and the frequency domain. In the time domain, the sequence is visually indistinguishable from white noise, but in the frequency domain, the frequency content becomes immediately clear.

After the sequences are generated, the next step is to use them to guide perturbations in the model. Consider region $A$, one of the regions to be perturbed, and also consider its corresponding sequence $\{z_i^A\}_{i=1}^{7300}$, where 7300 is the number of days in the 20-year simulation (CESM has 365 days in all years). Let $T_i^A$ be the temperature of the lowest model layer of region $A$ on day $i$. Then for each model day $i$, $T_i^A$ is replaced by $T_i^A + z_i^A$ at each model timestep in that day. This process is done simultaneously for all other regions that are being perturbed. We note that because the $\{z_i^A\}$ are uniform across each region, there will be discontinuities at the region boundaries, which could pose problems, particularly for spectral dynamical cores. Further research will need to be undertaken to reveal how this can best be handled; one possibility could be scale space smoothing methods (Marvel et al., 2013).

Of course, while "adding temperature" to a model layer is straightforward in a climate model, this procedure is unphysical. In physical terms, this can be thought of as adding a heat source to the model. If the maximum perturbation is 1 K, then the maximum amount of heat flux (W m$^{-2}$) added is

$$\Delta Q = \frac{1.0\text{K} \cdot c_p \cdot \rho \cdot h}{\tau} \qquad (9)$$

where $c_p$ is the specific heat capacity of air ($\sim 1000$ J kg$^{-1}$ K$^{-1}$), $\rho$ is the density of air ($\sim 1.2$ kg m$^{-3}$), $h$ is the height of the lowest model layer ($\sim 100$ m), and $\tau$ is the model timestep (1800 s). Because the perturbation is changed on a daily basis, the perturbation is the same for all model timesteps in a given day. By Equation 9, the maximum heat flux into any one region is approximately 67 W m$^{-2}$. This is a rather large perturbation over such an expansive region, but it is important to remember that the long-term mean of the perturbations over the course of the entire simulation is zero (Figure 3a), so to first order, there is no long-term net heat added to any one region or the climate system as a whole. This can be placed in context with a step response simulation in which there is a sustained 1 K increase in the lowest model layer over one region. This sustained temperature increase corresponds to approximately $3.4 \times 10^{22}$ J of added heat per year of simulation. Figure 4 shows a comparison between the inter-annual standard deviations of the preindustrial control run and the system identification ensemble. (By inter-annual standard deviation, we mean that the average over each year of simulation is used as an independent degree of freedom in the calculation.) Although we expect variability to be different between the two runs (the system identification perturbation is adding variability at a variety of frequencies), differences in standard deviations between the two simulations are negligible. This supports our claim that the perturbations added to the system identification simulations do not substantially alter the long-term climate.

All system identification simulation results subsequently presented are averages over an ensemble of five system identification simulations, for which five different sets of sequences were generated. Inter-ensemble variability is discussed in Section 3.4.

## 3.2 Steady State Response

Figure 5 provides an illustration that this method can recover some features the step response. The system identification panels (middle) were created by projecting (Equation 8) the entire time series of the output fields (low cloud fraction or latent heat flux) onto the sequence corresponding to a region in the Northwest Indian Ocean. The step response panels were calculated from an ensemble of five simulations in which, beginning from a preindustrial control run, the temperature in the lowest model layer over that region was instantaneously increased by 0.5 K, and that temperature change was sustained for 20 years. The maps displayed in the bottom panels of Figure 5 are twice (i.e., normalized to a perturbation of 1 K) an average over all 20 years of three ensemble members of that simulation minus an average over the preindustrial control simulation. As can be seen from this figure, the system identification simulation is different from the preindustrial control simulation (top row of Figure 5) and matches the broad features of the step response simulation quite well. There are differences between the step response and the system identification simulations, which could be due to the following:

1. The step response simulation involves adding approximately $1.7 \times 10^{22}$ J of heat to the climate system per year over 20 years (for a sustained 0.5 K perturbation), potentially exciting nonlin-

earities in the response (see Section 3.5 below), whereas to first order, the system identification simulation adds no net heat.

2. The step change (Equation 1) and the system identification inputs have different frequency contents and hence excite different responses on the timescales being analyzed in Figure 5. More specifically, the step response simulation is injecting a lot more energy at low frequencies than the system identification simulation, so the step response is in effect the low frequency response. Conversely, the system identification simulation injects a similar amount of

energy over a wide range of frequencies, so the resulting plot in Figure 5 is on average representative of the response at higher frequencies. As such, perfect agreement would not be expected.

## 3.3 Frequency-Dependent Response

As was stated previously, one of the advantages of this method (in addition to giving estimates for

all 22 regions simultaneously) is that it can characterize the input-output response dynamically (on many timescales) instead of only revealing the quasi-static response. Different relationships (e.g., local climate response or teleconnections) have different timescales on which different responses occur; by selectively band-pass filtering the signals when performing projections, one can isolate the climate response on specific timescales (as was discussed in Section 2.2).

As an example, Figure 6 shows the sensitivity of low cloud fraction to a 1 K temperature perturbation over the Northwest Indian ocean (the same region previously analyzed), calculated for different bands spanning approximately one-month timescales. The input-output relationships in Figure 6 appear to show the strongest signal on shorter timescales (although the shortest timescale that can be evaluated here is two weeks), with a peak response on the order of 1–2 months. The SNR declines

considerably as longer timescales are analyzed, and after a few months, there is no discernible signal beyond the noise. Figure 7 shows a similar picture for latent heat flux. This difficulty with ascertaining the signal from bands representing successively longer timescales is that the signal remains relatively constant with lower frequencies, whereas the "noise" (climate variability) increases with lower frequencies (not shown).

The results for sensitivities for band-pass filtering with a timescale of 1–2 months look quite similar to the steady-state response patterns in Figure 5. Figure 8 shows that including these early timescales as well as successively longer timescales does not affect how well the steady state response is recovered. (Figure 5 shows inclusion of the longest timescales that appear in the simulations.) This indicates that for the two variables evaluated here, the quasi-static response is reached

quite early in the simulations. This is consistent with the known rapidity of cloud and latent heat flux adjustments (examples of fast responses) to change (Cao et al., 2012). Such information is in principle evident in the step response simulations, although the signal only emerges above the noise when averaging the step response over a few years.

### 3.4 Statistical Significance

We performed two tests of statistical significance on our results. The first is to assess whether the results of the system identification simulations are distinguishable from noise, and the second is to assess inter-ensemble robustness of the results.

First, we generated 1000 sequences with the same characteristics as those described in Section 3.1 but are not mutually uncorrelated. We then projected (using all 7300 points in each sequence) the

300 preindustrial control simulation onto each sequence, forming a 1000-member ensemble of sensitivity maps. We then calculated the standard deviation across that ensemble to get an estimate of the range of values that might be expected from an unperturbed simulation, i.e., how large the impact of natural variability is on the system identification estimates. The responses estimated from system identification are more than two times larger than the standard deviation expected due to natural

variability.

For the second test, Figure 9 shows the standard deviation of the ensemble sensitivity (projections use all 7300 points in each sequence), where in calculating standard deviations, each of the five input sequences/ensemble members is considered an independent degree of freedom. Results show that there is somewhat more variability in the system identification ensemble than in the preindustrial

control simulation. Figure 10 shows the ensemble mean sensitivity values (repeated from the middle row of Figure 5) and those same fields but masked out where values are not statistically significant at the 95% confidence level according to a two-sample unpaired Student's $t$ test calculated on the inter-ensemble standard deviation (Figure 9). The results directly in the areas that are being perturbed are statistically significant, as are some far-field features in the midlatitudes.

### 3.5 Nonlinearity

As was mentioned previously, one of the potential sources of differences between the system identification and step response simulations is due to nonlinearities excited by the step response. To further explore these nonlinearities, we conducted two additional step response simulations involving perturbations over the Northwest Indian Ocean of $+0.2$K and $-0.5$K. The sensitivity maps (Figures 11

and 12) take the results of these simulations and divide by the perturbations to yield sensitivity maps that are comparable to those presented previously.

The results verify that the step response simulations do indeed introduce nonlinearities into the climate system. In the 0.2K simulation, there are many noisy features of climate response due to the lower signal-to-noise ratio inherent in that simulation as compared to the original 0.5K simulations.

We also note that the results presented for the 0.2K simulation will inherently be noisier than for the 0.5K simulations due to the difference in the number of ensemble members incorporated in the averages. The $-0.5$K simulation indicates substantial nonlinearities in the response in the form of

asymmetries. The 0.5K response appears to be stronger than the $-0.5$K response, although there are few locations that show prominent responses in one simulation but not the other.

These results suggest the need for a "gold standard" of the linear response to perturbations. Then the step response and system identification responses can be compared with that standard to ascertain the degree to which each simulation introduces nonlinearities. Such endeavors are beyond the scope of this paper, in particular because it would require an exploration to determine the methodology that is most appropriate for extracting the linear response. We discuss some potential methods in the following section.

## 4   Discussion and Conclusions

Here we have illustrated a method of characterizing dynamic climate system behavior in a computationally efficient way that does not strongly excite nonlinearities. All of the results presented were an average of three 20-year simulations in which 22 regions are perturbed simultaneously. If these relationships were discovered using step response simulations, the computational expense would be quite a bit greater, as computing the step response for $n$ regions requires $n$ simulations. However, there may still be reasons why the more computationally expensive approach of step change simulations might be conducted, particularly if one wishes to characterize nonlinear behavior.

Section 2.2 presented one method of generating sequences for the perturbations. Instead, one could design sequences that alternate pseudo-randomly between positive and negative perturbations of a fixed magnitude. These so-called *spread spectrum* techniques (Simon et al., 1994) are useful in situations where the inputs can only meaningfully accept binary values (e.g., the presence or absence of sea ice or snow cover).

The results in Section 3.3 revealed the importance of physical understanding in both choosing input signals and interpreting the results. The results indicated that low cloud fraction and latent heat flux respond to change rather rapidly; such information clearly would have been useful if the response time of these fields was not known. In retrospect, the energy input on timescales longer than a few months is wasted for the purpose of understanding these two variables. However, other variables operate on longer timescales, so input over such a wide band may still prove useful for analyses of other variables. If one knew a priori that they were interested in processes that occur over a specific range of timescales (e.g., the effects of Pacific sea surface temperature perturbations from El Niño on California rainfall), one could simply input white noise that is band-pass filtered in correspondence with those times. Our purpose here is to demonstrate this technique, which is widely applicable to a variety of input-output relationships, depending on the interests of the practitioner.

For example, in Figure 5, one can see synoptic scale sensitivity in latent heat flux in the midlatitude storm tracks. Based on this figure alone, and in the absence of a physical mechanism to cause such changes, it is difficult to say whether there are discernible responses to the input perturbation

or simply noise. However, the advantage of system identification is that it immediately provides one with tools to further investigate the potential for a response. Figure 7 further shows that the magnitude and even the sign of these features varies depending on the timescale in which one is interested. Analyzing the response to a different perturbed region (not shown) can help ascertain whether that response is particular to perturbations in a single region or whether this is the result of excitation of a natural mode of variability; in the latter case, information about the timescale of response can aid in identifying which mode of variability is being excited. In addition, one could isolate particular spatial areas that one wishes to analyze (for example, by spatial averaging over the midlatitudes) and compute the transfer function (MacMartin and Tziperman, 2014) to ascertain magnitude, phase, and spectral coherence of the relationship between that feature and the input signal. Through these explorations, one has a much greater chance of teasing out a physical mechanism that can explain the teleconnection seen in the results. Many of these possibilities are lost in step response simulations.

The results in Section 3.3 also revealed that a step response is not necessarily an ideal simulation to reveal the quasi-static response of these variables. The response is quasi-static at low frequencies, but noise increases with lower frequencies, meaning that as long as one is in the quasi-static regime, SNR is higher for higher frequencies. As such, the system identification simulation that is band-pass filtered over high frequencies can provide a "better" (less noisy) estimate of the sensitivity than the step response, which represents low frequencies. More specifically, due to contamination of the step response by nonlinearity and due to a lower signal-to-noise ratio, the system identification panels in Figure 5 better represent the steady state response than the step-change simulation. Note that this line of reasoning only works in this case because the steady state response establishes early in the simulation; other input-output relationships may require greater care in ascertaining the steady state response.

The present study is intended to introduce system identification to climate science through an example and has barely begun to reveal the potentials and limitations of system identification. The methodology appears to be effective (for certain variables) when 22 regions were perturbed with a fairly low amplitude input signal, but it likely would not work for 1000 regions, as the SNR would be too low (due to forcing over such a small area) to allow for meaningful detection of signals, and cross-talk between the regions would interfere too heavily with ascertaining quantitatively robust results. At the heart of this latter concern is nonlinearity. This method is based on linear theories and will not produce useful results for systems that are highly nonlinear (although the same is true of most methods, including step response simulations). The choice of boundaries between the regions may also have divided regions that potentially have physical connections. For example, in Figure 2, the Atlantic Ocean is covered by four regions, and no region spans the equator. This artificial introduction of an equatorial boundary would prevent identification of behavior in the Atlantic equatorial region. In principle, after separately identifying input-output relationships for the North and South Atlantic, we could add the two results to identify the response of the entire Atlantic basin, but this

might wash out more regional signals. Moreover, if the response to one input is positive and to another is negative, then the sum of these two responses may be small, masking sensitivities of smaller regions. These caveats are not indicative of potential failings in the approach, but in our application of it.

A point worth mentioning is the choice of input signal magnitude and how that may introduce concerns related to nonlinearities and the signal-to-noise ratio. In the present manuscript, we chose a maximum amplitude of the input signal to be 1K. This choice was somewhat arbitrary. Larger input signals will improve the detectability of the response but are also more likely to introduce nonlinearities. Smaller signals are less likely to introduce nonlinearities but will also have lower signal-to-noise ratios, making the response harder to determine. In addition, the spectra of responses will likely differ for different regions, so some regions may ultimately require different input signal magnitudes to achieve the same response confidence. An important future endeavor in establishing this system identification methodology will be to rigorously define and quantify both the signal-to-noise ratio and the degree of nonlinearity in the response. This will aid in determining the "optimal" magnitude of input signals.

Although system identification requires the assumption of linearity, the linear part of the response represents a substantial portion of the total response in a wide range of situations. Linear, time-invariant emulators, of which pattern scaling is a special case, show good fidelity to general circulation model simulations for a wide range of variables and forcings (e.g. Barnes and Barnes, 2015; Kravitz et al., 2016a; MacMartin and Kravitz, 2016; Santer et al., 1990). Other methods, such as Green's Function approaches (Hassanzadeh and Kuang, 2016) or application of the Fluctuation Dissipation Theorem (Leith, 1975; Gritsun and Branstator, 2007; Ring and Plumb, 2008; Cooper and Haynes, 2011; Fuchs et al., 2015) are other linear methods that have shown skill in recovering complex climate model behavior. Each of these methods has advantages and disadvantages; there is a great deal of promise in utilizing multiple complementary approaches to understand (linearized) input-output relationships in climate models. Also, as was briefly mentioned in Section 3.5, it is crucial to understand which situations are dominated by linear behavior versus which situations have a substantial nonlinear component to both understand the applicability of linear methods and to better quantify climate system nonlinearities.

The potential applications of this technique are numerous. Here we have briefly mentioned teleconnections; some specific examples include El Niño Southern Oscillation (ENSO) effects (e.g., Alexander et al., 2002) or propagation of the Madden-Julian Oscillation (e.g., Matthews, 2000; Gill, 1980). In particular, ENSO explorations (wherein the inputs could be changes in tropical Pacific sea surface temperatures) will be a useful future test of this method, as the ENSO cycle can be as long as 7 years, but responses can happen on the order of weeks to months (Alexander et al., 2002). However, exploring ENSO teleconnections would likely require inputs with different frequency content

than is used here. Our choice of white noise is the most agnostic choice, but as described previously, it is clearly not optimal if one has prior information about the dynamics of the system.

The method could also be used to explore the effects of marine cloud brightening to ascertain the optimal location to induce a perturbation (Latham et al., 2012), keeping in mind that model behavior is likely different from real-world behavior or even behavior in other models. Parkes (2012) showed preliminary results indicating that, with careful application, this method could be used to identify an "everywhere-to-everywhere transfer function" (S. Salter, personal communication) that fully characterizes the climate system response to marine cloud brightening in different regions. It could also be used to explore source-receptor relationships, which yield clearer and more quantitatively precise results but at the expense of computational cost. Moreover, these relationships are often uncovered via step response simulations. System identification could additionally be used in uncertainty quantification (UQ) studies to understand the climate response to perturbations in model tuning parameters. Current methods of UQ are quite expensive and involve step changes in tuning parameters, so the results of most UQ studies do not capture the full dynamic range of climate model response. This is not meant to be an exhaustive list, but merely an illustration of the sorts of problems where system identification may be useful.

## 5   Code and/or Data Availability

All model output and analysis code will be available upon request. Please contact the lead author to obtain this information.

*Acknowledgements.* We thank Daniel Kirk-Davidoff and one anonymous reviewer for their helpful suggestions in improving this manuscript. We thank Stephen Salter for bringing this concept to our attention and for his generosity in making time for repeated discussions. We also thank Hansi K. A. Singh, Susannah M. Burrows, and Jin-Ho Yoon for helpful discussions. This work was supported in part by the Regional and Global Climate Modeling program of the Office of Biological and Environmental Research in the United States Department of Energy's Office of Science as a contribution to the HiLAT project. The Pacific Northwest National Laboratory is operated for the U.S. Department of Energy by Battelle Memorial Institute under contract DE-AC05-76RL01830.

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

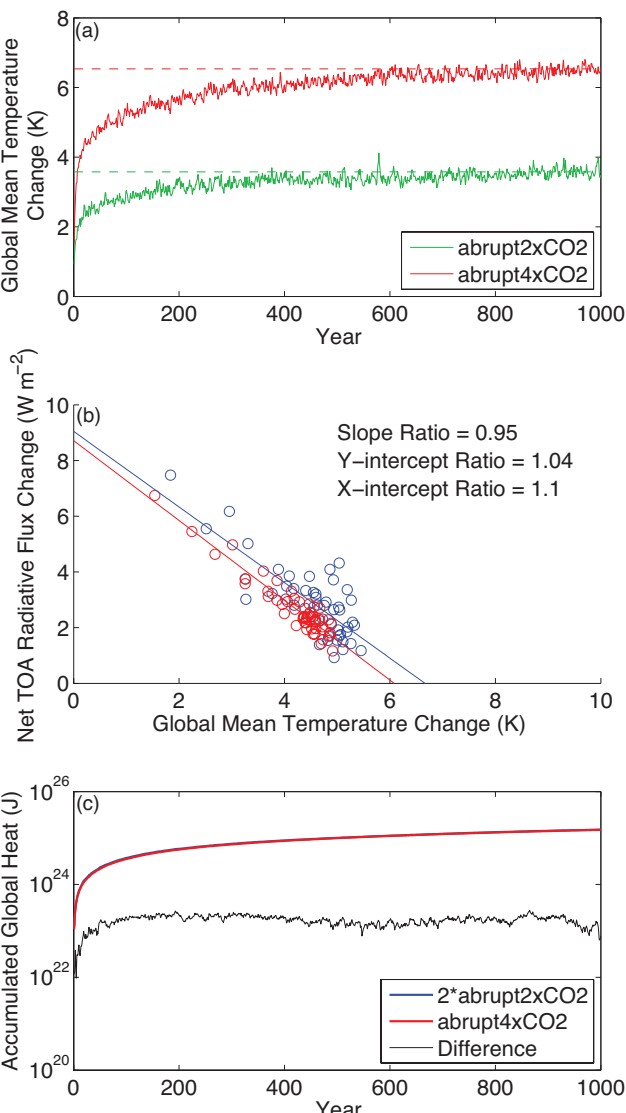

**Figure 1.** An illustration of nonlinearities in the climate system induced by step response simulations that, although not dominating climate system behavior, are potentially non-negligible. All simulations were conducted with the fully coupled general circulation model HadCM3L (Jones, 2003). Top panel shows timeseries of the change in global mean temperature in abrupt2xCO2 (green) and abrupt4xCO2 (red) simulations; approximate steady state values are indicated by dashed lines. Middle panel shows annual mean temperature change and top-of-atmosphere (TOA) net radiative flux differences ($\Delta R$) from a preindustrial control (circles) for the first 50 years of twice the abrupt2xCO2 simulation (blue) and the abrupt4xCO2 simulation (red); lines are ordinary least squares regression through the respective circles. Bottom panel shows approximate global heat uptake for twice the abrupt2xCO2 simulation (blue) and the abrupt4xCO2 simulation (red) calculated as in Equation 2; black line shows the difference between the blue and red lines.

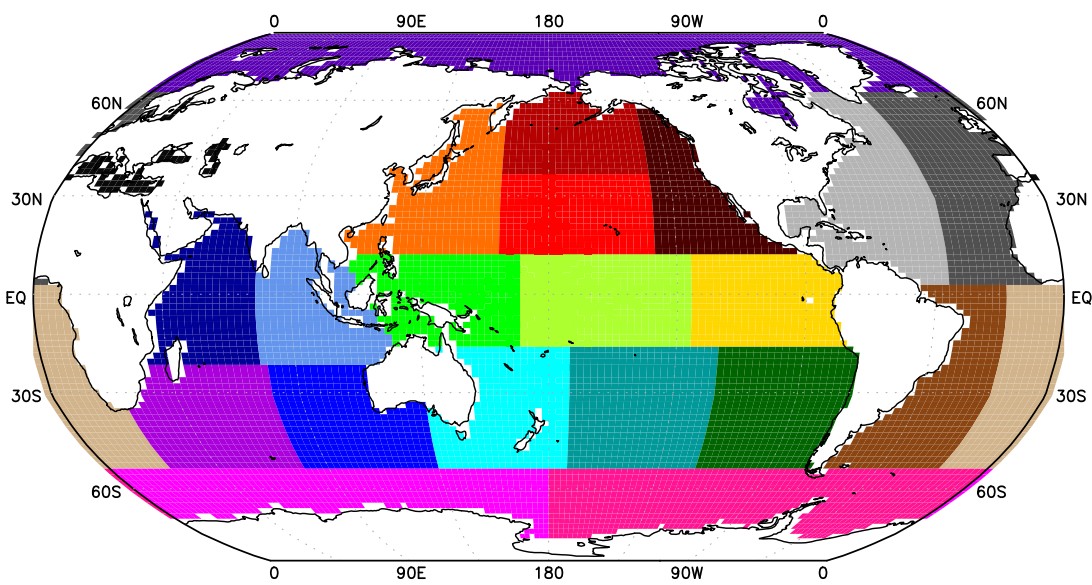

**Figure 2.** The 22 regions that were perturbed (see Section 3.1) in this study. Regions are approximately equal in area, and no region spans multiple ocean basins.

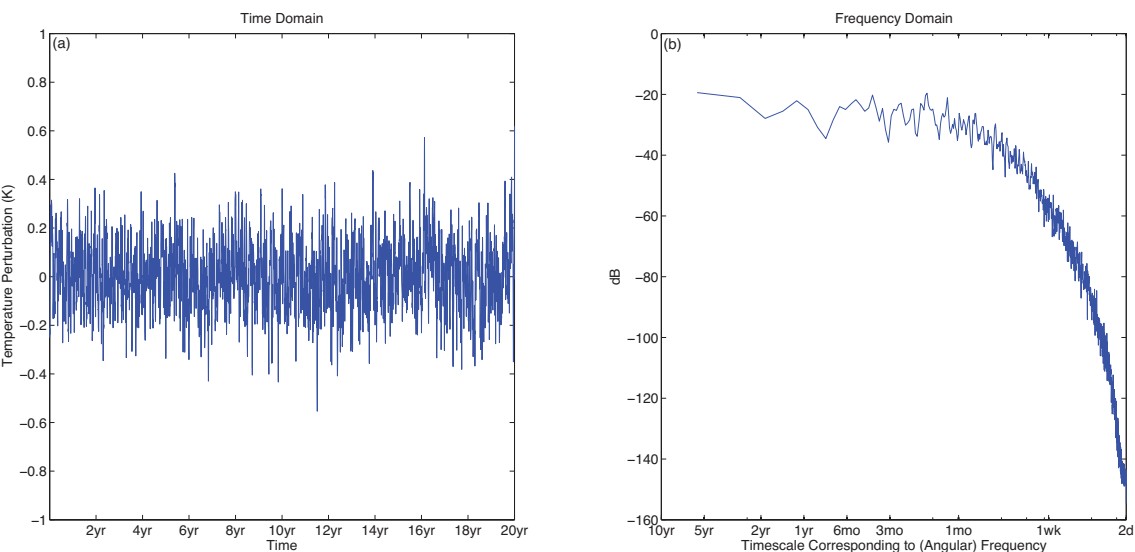

**Figure 3.** Time domain (left) and frequency domain (right) representations of one of the 22 sequences used in this study to perturb temperature (see Section 3.1). The sequences are low-pass filtered white noise with a cutoff frequency corresponding to a timescale of one week.

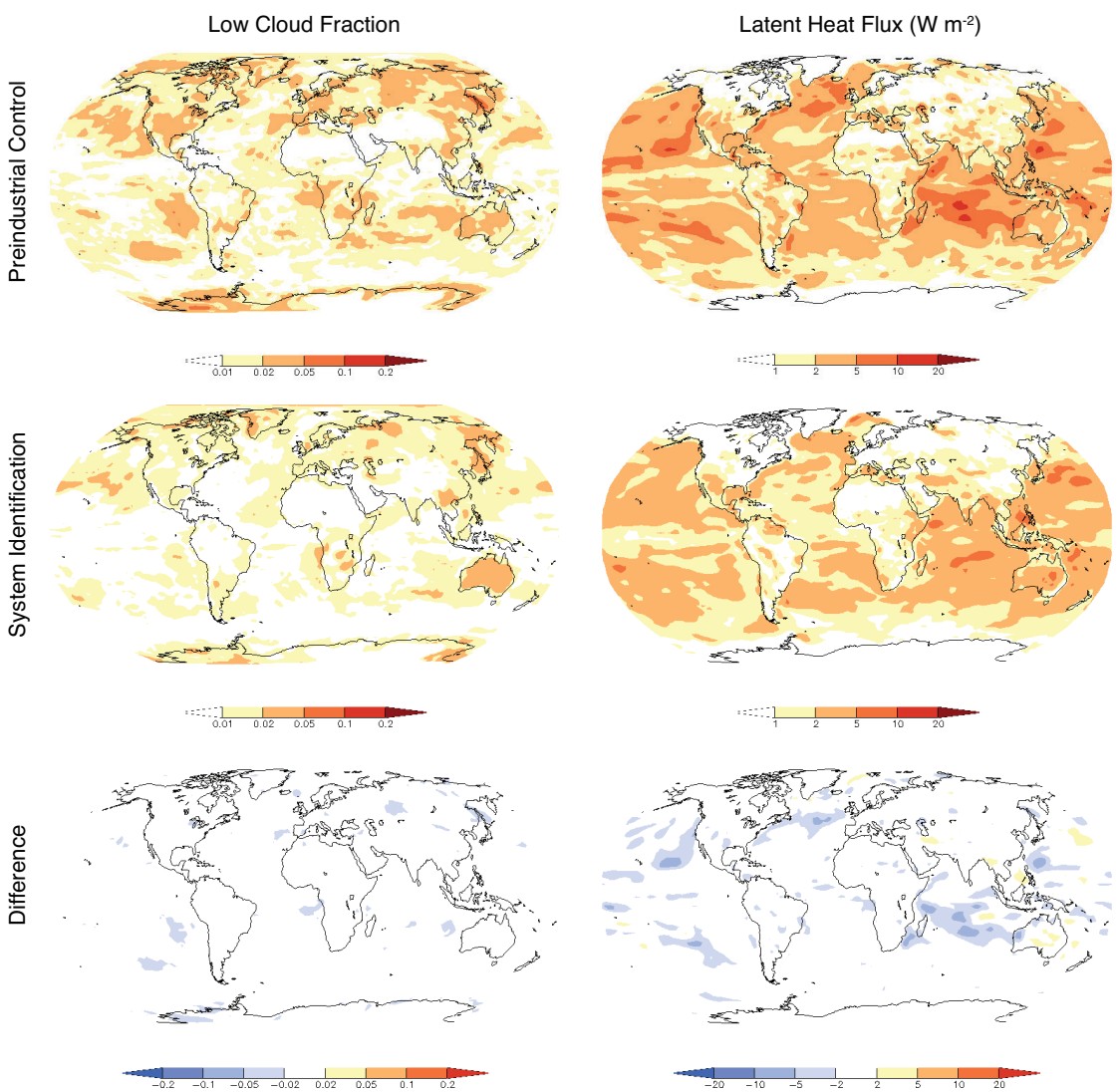

**Figure 4.** Inter-annual standard deviation for the simulations considered here. Values are calculated using the annual mean maps as independent degrees of freedom. The preindustrial control values are calculated using a single 40-year simulation (39 degrees of freedom). The system identification values are calculated using a five-member ensemble of 20-year simulations (95 degrees of freedom). Differences are the middle panels minus the top panels.

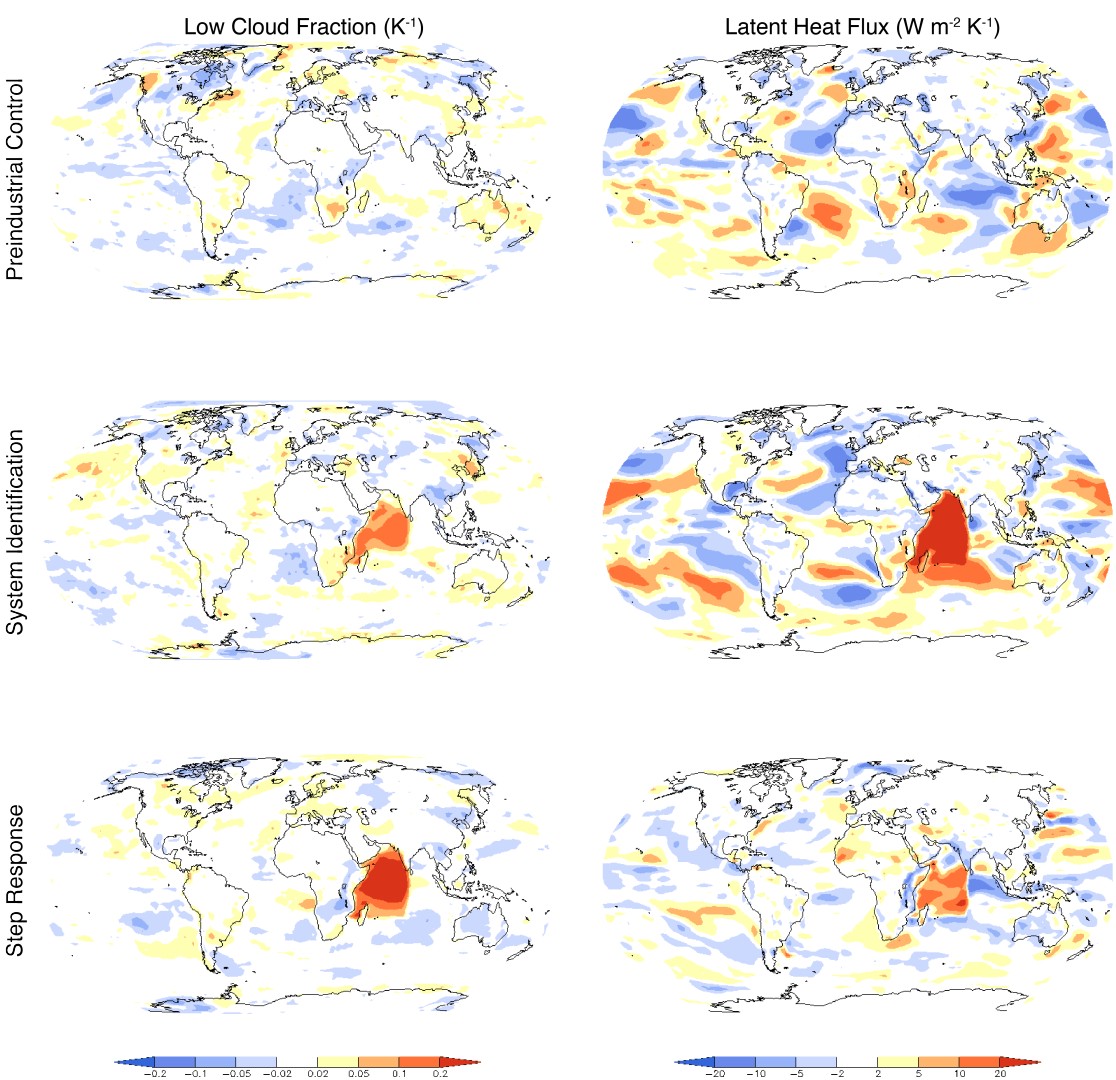

**Figure 5.** Steady state response of low cloud fraction (left column) and latent heat flux (right column) for a 1 K perturbation to the lowest model layer over the Northwest Indian ocean. Top row shows projections of the unperturbed preindustrial control simulation onto the input sequences; no response beyond climate system noise is expected. Middle row shows projections of the system identification (perturbed) simulations onto the input sequences (all 20 years of simulation). For comparison, the bottom row shows step response simulations in which the highlighted region has a sustained temperature increase over the 20 year simulation (values shown are averages over the entire 20-year period). Although somewhat noisy, the system identification simulations are capable of recovering the broad features of the step response.

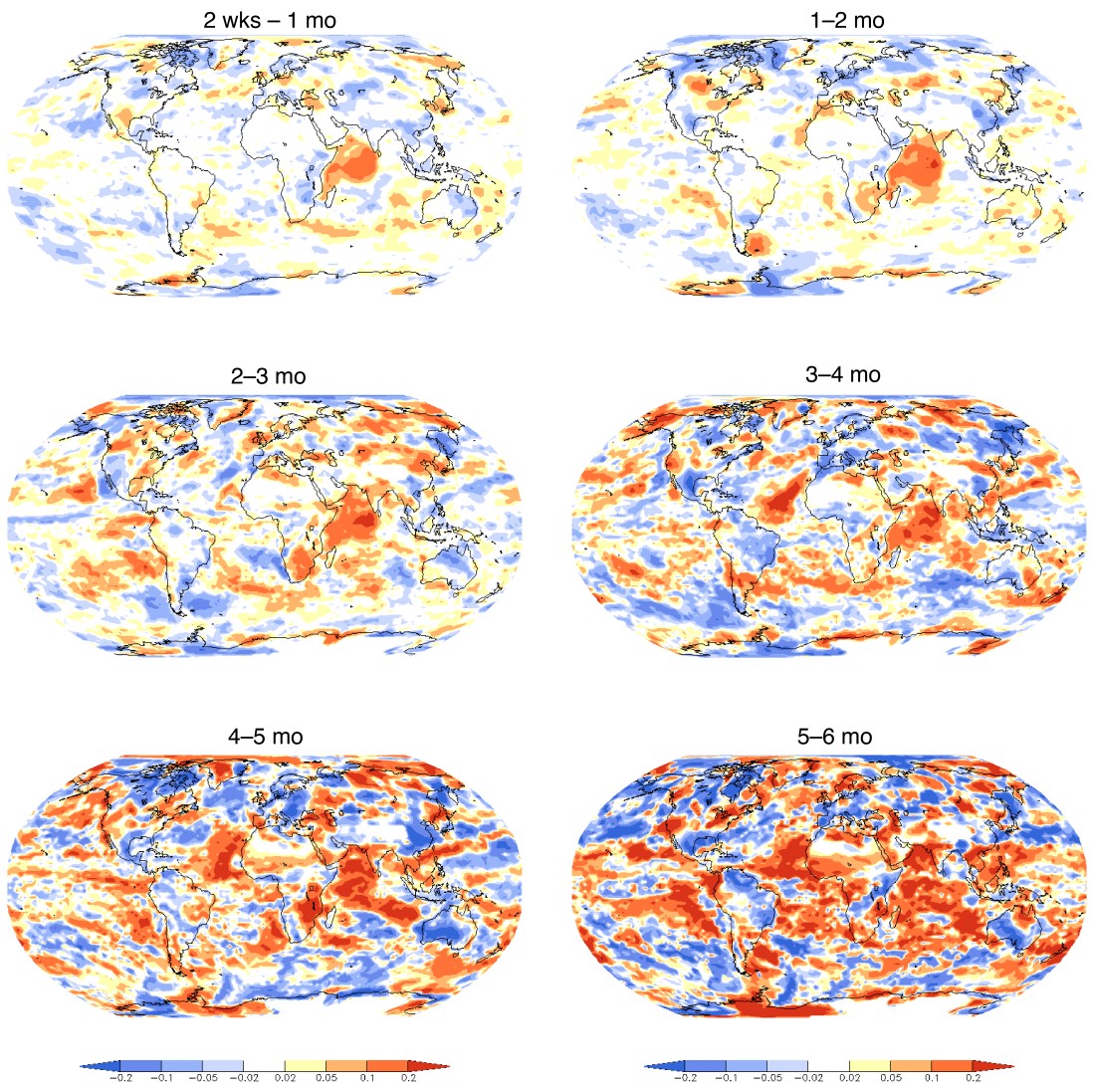

**Figure 6.** Sensitivity of low cloud fraction to a 1 K temperature perturbation to the Northwest Indian Ocean (see Figure 2). Different panels were calculated from projections on band-pass filtered timeseries (see Section 3.3).

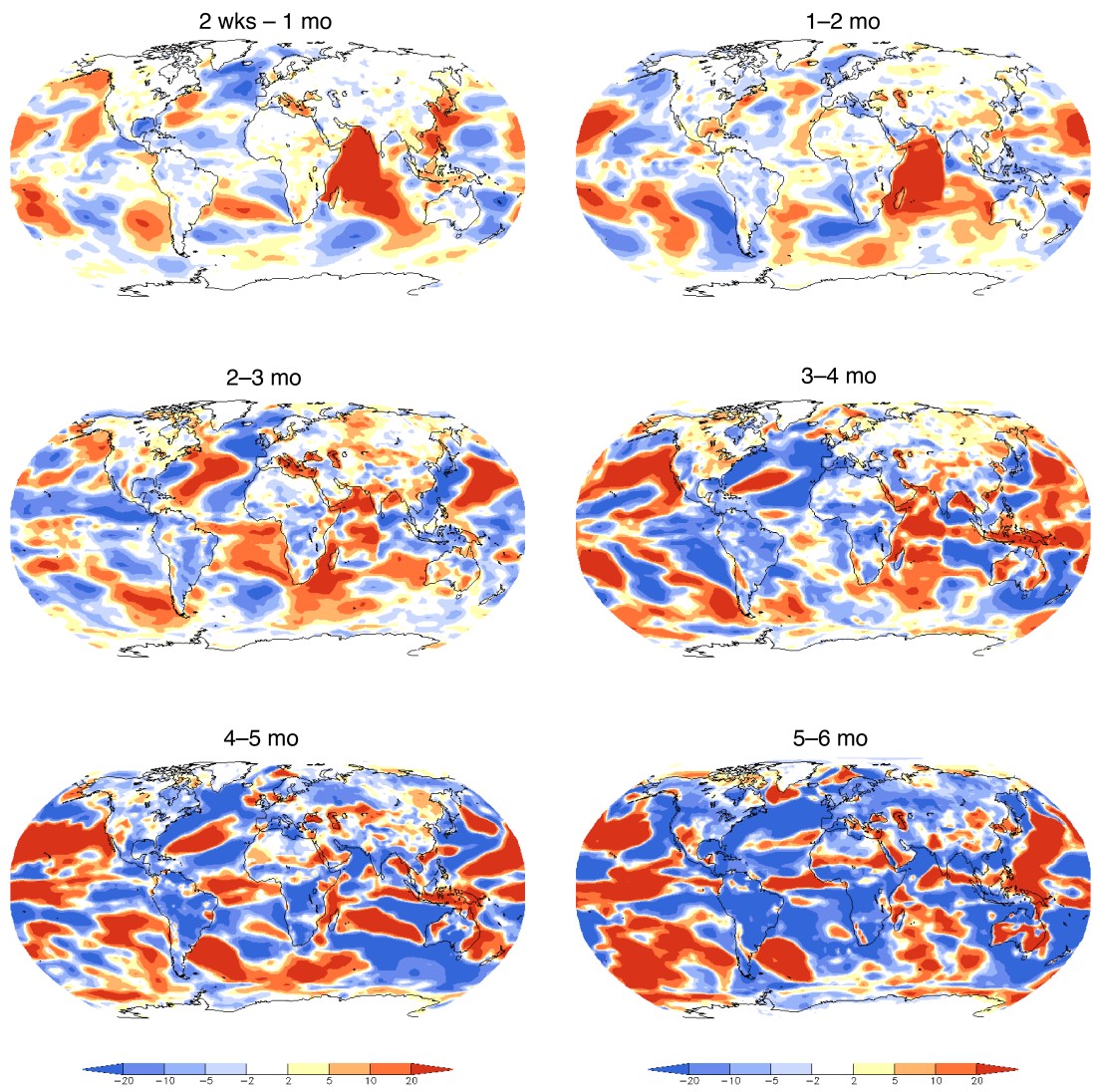

**Figure 7.** As in Figure 6 but for the sensitivity of latent heat flux changes to a 1 K temperature perturbation to the Northwest Indian Ocean.

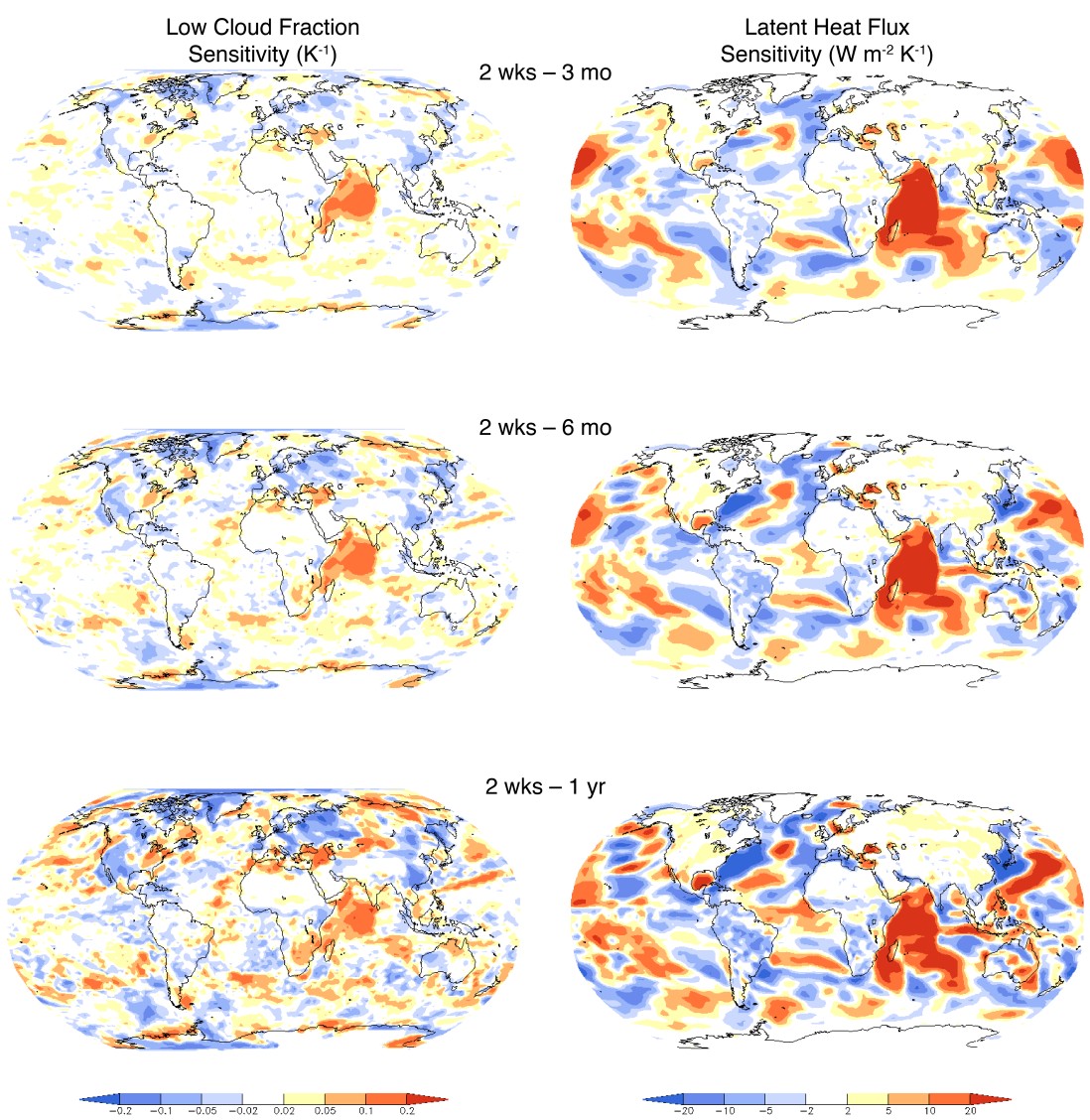

**Figure 8.** As in Figures 6 and 7 but for bands including wider ranges of frequencies.

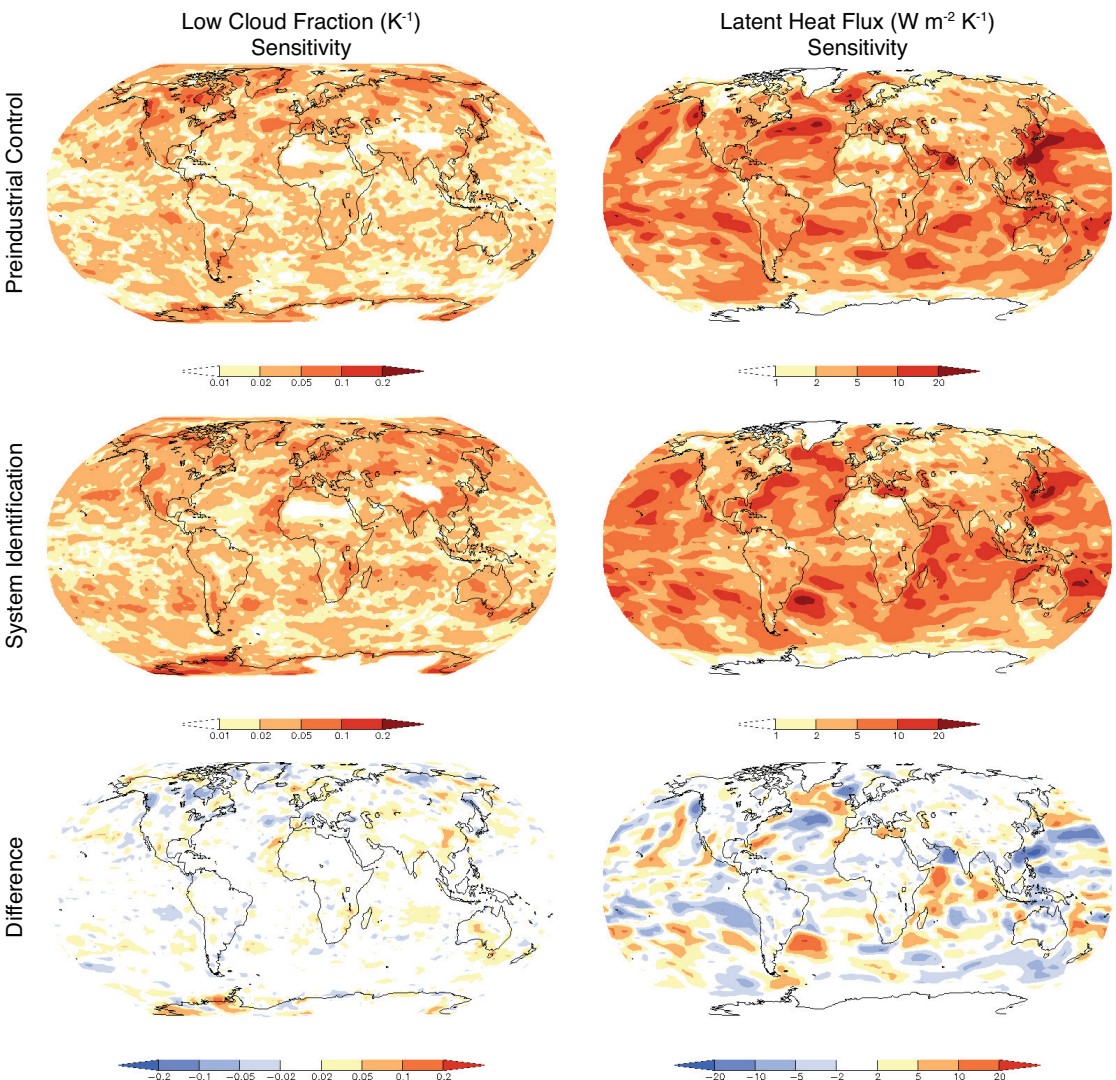

**Figure 9.** Inter-ensemble standard deviations of the sensitivities of low cloud fraction and latent heat flux. Sensitivities are calculated via projection onto the full sequences that are 7300 days in length. For the control simulation, ensemble members were generated by projecting the control run onto each of the five sequences considered here. Differences are the system identification inter-ensemble standard deviation (middle panels) minus the control inter-ensemble standard deviation (top panels).

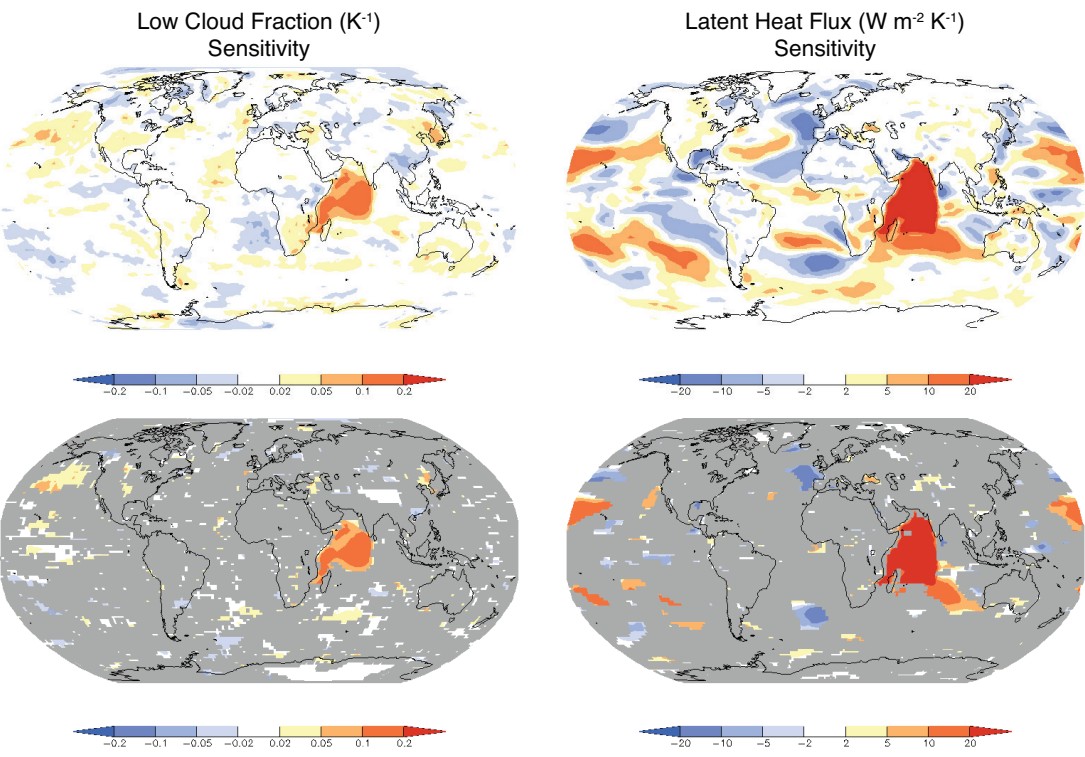

**Figure 10.** Top row shows sensitivities calculated by projection over the entire 7300 day simulation (repeated from the middle panels of Figure 5. Bottom panels show the same values but masked out (grey) where they are not statistically significant at the 95% confidence level (two-sample unpaired Student's $t$ test) as calculated from the standard deviation values presented in Figure 9.

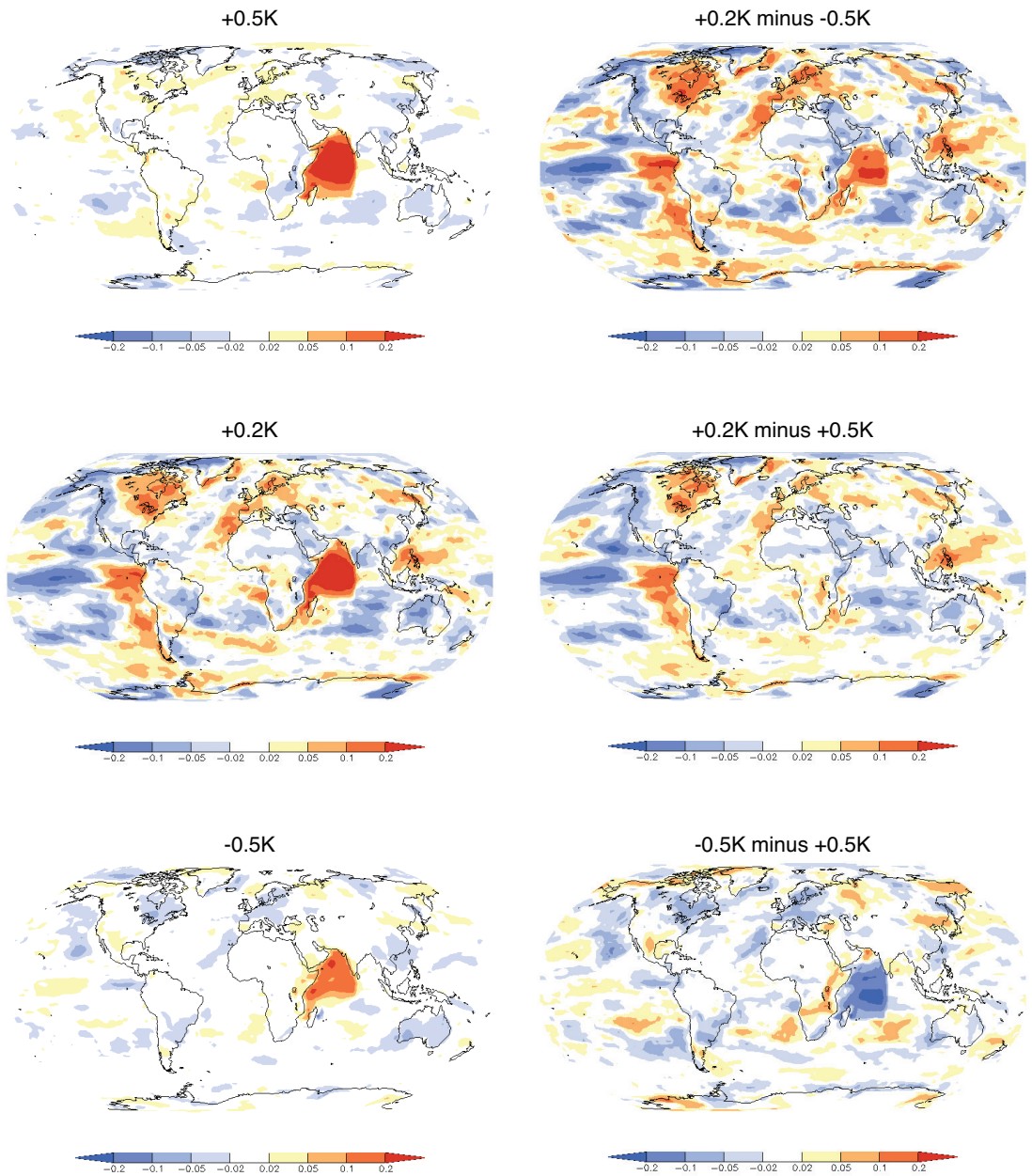

**Figure 11.** Sensitivity (left column) and differences in sensitivity (right column) of low cloud fraction to different magnitudes of step change. All values are in units of $K^{-1}$. Top left shows the sensitivity to a sustained increase in lower atmospheric temperature by 0.5K (as in previous figures). Middle left and bottom left show sensitivity to sustained lower atmospheric temperature changes of 0.2K and $-0.5$K, respectively. These are calculated by conducting simulations in which heat is added or subtracted accordingly, and then the results are normalized by the perturbation. The 0.5K simulation results are for an average of five ensemble members; other simulation results are for single ensemble members.

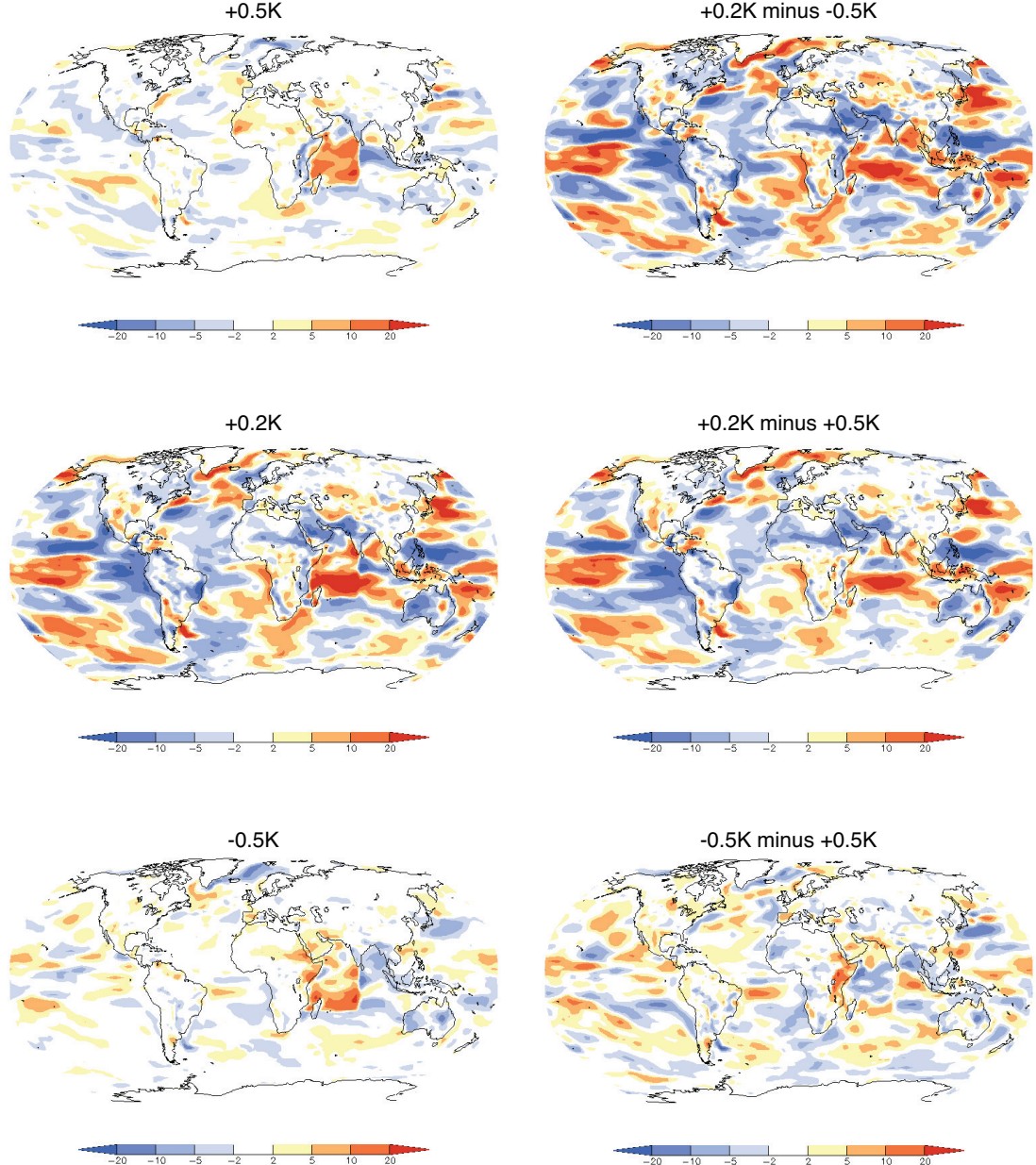

**Figure 12.** As in Figure 11 but for latent heat flux sensitivity (W m$^{-2}$ K$^{-1}$).