# Peer review of "Technical Note: Simultaneous fully dynamic characterization of multiple input-output relationships in climate models"

_Atmospheric Chemistry and Physics, 2016_

## Referee Comment (RC1) · D.B. Kirk-Davidoff (Referee) · 18 Sep 2016

In this paper, Kravitz and co-authors introduce the System Identification (SI) technique to probe the linear dynamical response of climate models to localized perturbations. They demonstrate that by perturbing the temperature of adjacent regions of the ocean with orthogonal noise and then filtering the global response by the the time series for each region, they can approximately reproduce the effect of a step change in temperature in one of those regions on the low cloud and surface latent heat flux fields.

While I found the approach intriguing, I feel that there are three key analysis steps missing from a publishable paper. First, since the study reported here involved three simulations using the SI procedure, it seems odd that only ensemble means, but no

ensemble variability was shown. The results were said to be robust across the three runs, but why not show the readers the variability? Why not perform a few more runs, so that some measure of statistical significance of the results could be shown, for example by stippling, in the figures?

Second, there's little discussion of the character of the differences between the step-change runs and the SI results. What's going on with the synoptical scale variability in latent heat flux found in the mid-latitude storm tracks in SI runs (figure 4, middle row, right column)? Are these natural modes of variability whose frequency happens to correspond to the frequencies excited over the Indian Ocean? Are these patterns also found in the response for many other regions. In this regard, it would be helpful to see the response patterns to the perturbation introduced in at least one other region besides the Indian Ocean, even without a corresponding step-change experiment. Are the remote responses corresponding to the perturbations in these regions similar to the ones from the Northwest Indian Ocean? If so, is there a filtering procedure that could potentially be used to screen them out?

Finally I believe it would be helpful if the authors could clarify the usefulness of the detection of purely linear responses to climate perturbations (subject to some accidental contamination with non-linear responses). Since the response to any observable forcing will include non-linear features, the authors should say a little more about the benefits and hazards of isolating the linear response using SI.

---

## Referee Comment (RC2) · Anonymous Referee #2 · 13 Oct 2016

**Review**

The paper discusses applying a system identification technique to find the responses of climate models to small perturbations using a single simulation. The objectives and the methodology are well described in sections 1 and 2. The method is then applied in section 3 to find how low cloud fraction and latent heat flux change globally in response to a +1K perturbation of the near-surface temperature over the Northwest Indian Ocean. The results are compared with the results of a step response simulation. The results and the potential applications of the technique are discussed in section 4.

I find the technique very interesting and I believe that it can be a powerful tool in studying the climate system and that it has various potential applications (as some described in section 4). However, the manuscript should be first improved in several ways before it is considered for publication:

1) It helps the reader and potential users of the technique if the paper presents a quantitative evaluation of how well the proposed technique works in the single example shown

2) Further discussions on how the range of the magnitude of the normalized sequences is chosen (Line 200) and how SNR and nonlinearity can be quantified are needed.

3) There are few other techniques proposed in the literature to characterize the linear response of climate models. They should be briefly mentioned in section 1.

Here are some details regarding 1-3:

1) Fig. 1 shows that the system identification technique underestimates (overestimates) the magnitude of the low cloud fraction (latent heat flux) response. The patterns of the response, which are simple and local, look similar. The errors in the magnitude and pattern (e.g. pattern correlation in the region of interest) should be quantified. That way how changing parameters such as the frequencies of the band-pass filtering affects the accuracy can be quantified. Among the two reasons offered for the discrepancies, contribution from nonlinearity can be eliminated by reducing the amplitude of the forcing in the step function simulation and using a pair of heating and cooling simulations. Finding the "true" linear response using the step function simulations against which the system identification technique can be verified might take some effort, but given that evaluating the performance of the technique is crucial, I think it is worth the effort to demonstrate how well the technique captures the linear response. As for the second reason, again, it would be nice to quantitatively show which range of band-pass filtering yields the best agreement with the linear response of the step function simulation at least for this specific example. I am also wondering why the response to this specific region (Northwest Indian Ocean) and in these specific variables have been chosen.

2) How is the [-1K,+1K] range is chosen for the normalized sequence? (Line 200). In problems concerning linear responses, a major difficulty is obtaining good SNR without violating linearity. Does this problem exist in this technique as well? If so, how the SNR and nonlinearity in depend on the magnitude of this range should be explored (and this likely vary across different regions), which requires quantifying the SNR and degree of nonlinearity. I am not suggesting that this should be done for this paper, but these issues should be (at least briefly) discussed.

3) Other methods have been proposed in the literature which share closely some of the objectives of the current study. One example is the fluctuation-dissipation theorem (some recent examples: Gritsun and

Branstator 2007, Ring and Plumb 2008, Cooper and Haynes 2011, Fuchs et al 2015). Another example is the Greens function method of Hassnazadeh and Kuang 2016. Note that these methods too can determine (at least theoretically) the full dynamic behavior of the system as they provide the linear response function of the model. I suggest to briefly mention other methods such as these two to better connect the current paper with the literature.

Gritsun, A. and Branstator, G., 2007. Climate response using a three-dimensional operator based on the fluctuation-dissipation theorem. Journal of the Atmospheric Sciences

Ring, M.J. and Plumb, R.A., 2008. The response of a simplified GCM to axisymmetric forcings: Applicability of the fluctuation-dissipation theorem. Journal of the Atmospheric Sciences

Cooper, F.C. and Haynes, P.H., 2011. Climate sensitivity via a nonparametric fluctuation-dissipation theorem. Journal of the Atmospheric Sciences

Fuchs, D., Sherwood, S. and Hernandez, D., 2015. An exploration of multivariate fluctuation dissipation operators and their response to sea surface temperature perturbations. Journal of the Atmospheric Sciences

Hassanzadeh, P. and Kuang, Z., 2016. The linear response function of an idealized atmosphere. Part 1: Construction using Green's functions and applications. Journal of the Atmospheric Sciences

**Minor comments:**

Simulation details: it helps if some details about the model setup (e.g. horizontal and vertical resolutions in the atmosphere and ocean etc.) are included following line 178.

End of line 264: ")" missing

Line 315: Figure's label missing

Fig 1a: dashed lines are visible on the pdf but do not show when printed

Line 205-215: $z^A_i$ is uniform across each of the 22 regions (right?). But that means there would be discontinuities in the heat source profile at the boundaries of these regions. That might cause problems if the model uses spectral methods.

Is there a statistically significant difference between the climatology (eg. time mean latent heat flux or zonal wind field) of the pre-industrial control simulation and the climatology of the system identification (perturbed) simulation?

I am very curious to see how this interesting technique performs for a range of problems and especially for regions that excite teleconnection patterns, as the authors discussed in section 4. To evaluate the technique, it might be a good idea (for future) to use a simpler climate model (e.g. an atmospheric GCM with coarse grid) which is cheaper to run and can be used to explore more examples.

---

## Author Comment (AC1) · 20 Dec 2016

Response to reviewers
ACP-2016-653
Original reviewer comments in normal typeface.  **Responses in bold.**
* * *
Reviewer #1:  Daniel Kirk-Davidoff

In this paper, Kravitz and co-authors introduce the System Identification (SI) technique to probe the linear dynamical response of climate models to localized perturbations. They demonstrate that by perturbing the temperature of adjacent regions of the ocean with orthogonal noise and then filtering the global response by the time series for each region, they can approximately reproduce the effect of a step change in temperature in one of those regions on the low cloud and surface latent heat flux fields.

**We thank the reviewer for the careful attention to our manuscript.**

While I found the approach intriguing, I feel that there are three key analysis steps missing from a publishable paper. First, since the study reported here involved three simulations using the SI procedure, it seems odd that only ensemble means, but no ensemble variability was shown. The results were said to be robust across the three runs, but why not show the readers the variability? Why not perform a few more runs, so that some measure of statistical significance of the results could be shown, for example by stippling, in the figures?

**The reviewer brings up a couple of valuable points.  We have removed these lines and added a new section in which we discuss this more quantitatively.**

**First, we conducted two additional system identification simulations; all results now show the ensemble average of five runs.  The addition of these runs did not change broad features of the sensitivity maps shown in the manuscript.  Moving from three to five ensemble members reduces the noise by a factor of sqrt(3/5)=0.77, so the plots should not be visually different.  One might expect differences in estimates of the inter-ensemble standard deviation, which we have now included in the manuscript as a figure.**

**To address the problem of statistical significance, we have now included a figure showing the ensemble-mean sensitivity, masked out where the inter-ensemble variability failed a Student's _t_ test at the 95% confidence level.  We performed a different method of calculating statistical significance by creating 1000 random sequences with the same frequency content as the other sequences used in the paper (although this time they are not mutually uncorrelated), projected the control simulation onto each sequence, and calculated the standard deviation across all 1000 projections.  This gives us a means of estimating natural variability in the sensitivity fields.  All values in the ensemble mean sensitivity field were over 2 standard deviations of the natural variability.**

**We have included a discussion of all of these new results in the revised manuscript.**

Second, there's little discussion of the character of the differences between the step-change runs and the SI results. What's going on with the synoptical scale variability in latent heat flux found in the mid-latitude storm tracks in SI runs (figure 4, middle row, right column)? Are these natural modes of variability whose frequency happens to correspond to the frequencies excited over the Indian Ocean? Are these patterns also found in the response for many other regions. In this regard, it would be helpful to see the response patterns to the perturbation introduced in at least one other region besides the Indian Ocean, even without a corresponding step-change experiment. Are the remote responses corresponding to the perturbations in these regions similar to the ones from the Northwest Indian Ocean? If so, is there a filtering procedure that could potentially be used to screen them out?

**This is a really useful comment.  We note that these values are not statistically significant using the metric presented in one of the new figures, so we are reluctant to discuss those features without strong evidence for a physical mechanism underpinning those results.  Moreover, because our intention with this manuscript is that the scope should remain within that of a Technical Note, we have eschewed detailed explorations of particular features.**

**Nevertheless, the reviewer brings up an excellent point, in that we have not fully described how the advantages provided by system identification could be used to gain deeper insight into teleconnections and their underlying physical mechanisms. We have added an additional paragraph to the discussion section describing these procedures, using the suggested mid-latitude storm tracks as an example.**

Finally I believe it would be helpful if the authors could clarify the usefulness of the detection of purely linear responses to climate perturbations (subject to some accidental contamination with non-linear responses). Since the response to any observable forcing will include non-linear features, the authors should say a little more about the benefits and hazards of isolating the linear response using SI.

**Agreed.  We have combined this with a suggestion from the other reviewer to include a more thorough discussion of other linear methods.  We have added an additional paragraph to the manuscript that addresses these issues.**

---

## Author Comment (AC2) · 20 Dec 2016

Response to reviewers
ACP-2016-653
Original reviewer comments in normal typeface.  **Responses in bold.**
* * *
Anonymous Reviewer #2

The paper discusses applying a system identification technique to find the responses of climate models to small perturbations using a single simulation. The objectives and the methodology are well described in sections 1 and 2. The method is then applied in section 3 to find how low cloud fraction and latent heat flux change globally in response to a +1K perturbation of the near-surface temperature over the Northwest Indian Ocean. The results are compared with the results of a step response simulation. The results and the potential applications of the technique are discussed in section 4.

I find the technique very interesting and I believe that it can be a powerful tool in studying the climate system and that it has various potential applications (as some described in section 4). However, the manuscript should be first improved in several ways before it is considered for publication:
1) It helps the reader and potential users of the technique if the paper presents a quantitative evaluation of how well the proposed technique works in the single example shown
2) Further discussions on how the range of the magnitude of the normalized sequences is chosen (Line 200) and how SNR and nonlinearity can be quantified are needed.
3) There are few other techniques proposed in the literature to characterize the linear response of climate models. They should be briefly mentioned in section 1.

**We thank the reviewer for the careful consideration of our manuscript and the excellent comments. We address each of these points below after the details are elucidated.**

Here are some details regarding 1-3:
1) Fig. 1 shows that the system identification technique underestimates (overestimates) the magnitude of the low cloud fraction (latent heat flux) response. The patterns of the response, which are simple and local, look similar. The errors in the magnitude and pattern (e.g. pattern correlation in the region of interest) should be quantified. That way how changing parameters such as the frequencies of the band- pass filtering affects the accuracy can be quantified. Among the two reasons offered for the discrepancies, contribution from nonlinearity can be eliminated by reducing the amplitude of the forcing in the step function simulation and using a pair of heating and cooling simulations. Finding the "true" linear response using the step function simulations against which the system identification technique can be verified might take some effort, but given that evaluating the performance of the technique is crucial, I think it is worth the effort to demonstrate how well the technique captures the linear response. As for the second reason, again, it would be nice to quantitatively show which range of band-pass filtering yields the best agreement with the linear response of the step function simulation at least

for this specific example. I am also wondering why the response to this specific region (Northwest Indian Ocean) and in these specific variables have been chosen.

**We like the reviewer's suggestions and have included the results from two additional step response simulations to better explore the role that nonlinearity plays in the results. After conducting these simulations, we found that extracting the "true" linear response is surprisingly challenging. There are sufficient nonlinearities in the step response that quite a bit of additional work is required (indeed, this would probably be best as the subject of a separate paper) and would require methods that are beyond the scope of what we can explore in the present manuscript. We have added an additional Section (3.5) and two additional figures that provide the results of these simulations, as well as additional description that provides explanation of the difficulties in extracting this linear signal. We have included a description of several paths toward properly validating the system identification method. We thank the reviewer for identifying what our next paper will be about!**

**As for the region and variables, our choice was somewhat arbitrary. After we conducted our initial simulations, we looked at the results for a variety of regions and variables, identified some interesting features that were illustrative of our purpose (to explain the methodology of system identification), and showed them. The simulations certainly contain other interesting results that would likely work equally well for our purposes, but as this is primarily a Technical Note describing the methodology, we did not want to distract from our primary purpose.**

2) How is the [-1K,+1K] range is chosen for the normalized sequence? (Line 200). In problems concerning linear responses, a major difficulty is obtaining good SNR without violating linearity. Does this problem exist in this technique as well? If so, how the SNR and nonlinearity in depend on the magnitude of this range should be explored (and this likely vary across different regions), which requires quantifying the SNR and degree of nonlinearity. I am not suggesting that this should be done for this paper, but these issues should be (at least briefly) discussed.

**The reviewer brings up several valuable points that we agree need to be discussed. We have added additional text to the conclusions section that addresses these concerns.**

3) Other methods have been proposed in the literature which share closely some of the objectives of the current study. One example is the fluctuation-dissipation theorem (some recent examples: Gritsun and Branstator 2007, Ring and Plumb 2008, Cooper and Haynes 2011, Fuchs et al 2015). Another example is the Greens function method of Hassnazadeh and Kuang 2016. Note that these methods too can determine (at least theoretically) the full dynamic behavior of the system as they provide the linear response function of the model. I suggest to briefly mention other methods such as these two to better connect the current paper with the literature.

Gritsun, A. and Branstator, G., 2007. Climate response using a three-dimensional operator based on the fluctuation-dissipation theorem. Journal of the Atmospheric Sciences

Ring, M.J. and Plumb, R.A., 2008. The response of a simplified GCM to axisymmetric forcings: Applicability of the fluctuation-dissipation theorem. Journal of the Atmospheric Sciences

Cooper, F.C. and Haynes, P.H., 2011. Climate sensitivity via a nonparametric fluctuation-dissipation theorem. Journal of the Atmospheric Sciences

Fuchs, D., Sherwood, S. and Hernandez, D., 2015. An exploration of multivariate fluctuation dissipation operators and their response to sea surface temperature perturbations. Journal of the Atmospheric Sciences

Hassanzadeh, P. and Kuang, Z., 2016. The linear response function of an idealized atmosphere. Part 1: Construction using Green's functions and applications.

**We thank the reviewer for pointing this out.  We have now included additional discussion in the final section that puts this method in the broader context of linear methods to understand climate model behavior.**

Minor comments:
Simulation details: it helps if some details about the model setup (e.g. horizontal and vertical resolutions in the atmosphere and ocean etc.) are included following line 178.

**Agreed.  Added.**

End of line 264: ")" missing Line 315: Figure's label missing Fig 1a: dashed lines are visible on the pdf but do not show when printed

**Thanks for the careful reading.  We'll alert the production staff of the journal.**

Line 205-215: $z^A_i$ is uniform across each of the 22 regions (right?). But that means there would be discontinuities in the heat source profile at the boundaries of these regions. That might cause problems if the model uses spectral methods.

**Yes, $z_i^A$ are uniform across the 22 regions.  We thank the reviewer for alerting us to the possible concern regarding spectral methods.  We have added a mention of this to the manuscript.**

**We note that the reviewer raises an interesting, broader issue in climate model simulations – for example, consider the land-ocean discontinuities introduced by conducting fixed sea surface temperature simulations.  Our familiarity with how climate model numerics behave in this case is insufficient for us to comment on it in the manuscript, but we are interested in exploring how this issue has been dealt with**

**in spectral models in the past, which could provide some insight into how our results may have been affected by discontinuities.**

Is there a statistically significant difference between the climatology (eg. time mean latent heat flux or zonal wind field) of the pre-industrial control simulation and the climatology of the system identification (perturbed) simulation?

**This is a good point. Please see the response to Reviewer #1 for a further discussion of statistical significance and how we have modified the manuscript to address this issue. We have also added a figure comparing inter-annual standard deviations of the climatology and perturbed simulations using each year as an independent degree of freedom (although we of course acknowledge that consecutive years may not actually be independent).**

I am very curious to see how this interesting technique performs for a range of problems and especially for regions that excite teleconnection patterns, as the authors discussed in section 4. To evaluate the technique, it might be a good idea (for future) to use a simpler climate model (e.g. an atmospheric GCM with coarse grid) which is cheaper to run and can be used to explore more examples.

**Great idea! We'll keep that in mind when we write our next paper.**

---

## Author Response (AR2)

We have made two technical corrections (adding units on Page 4), and we have added funding acknowledgments.  There are no other changes from the previous version.